

# The role of autophagy in high-fat diet-induced insulin resistance of adipose tissues in mice

Yovita Permata Budi[1,2,*], Yi-Hsuan Li[3,*], Chien Huang[3], Mu-En Wang[4], Yi-Chun Lin[5], De-Shien Jong[3], Chih-Hsien Chiu[3] and Yi-Fan Jiang[1,2]

[1] Graduate Institute of Molecular and Comparative Pathobiology, National Taiwan University, Taipei, Taiwan
[2] School of Veterinary Medicine, National Taiwan University, Taipei, Taiwan
[3] Department of Animal Science and Technology, National Taiwan University, Taipei, Taiwan
[4] Department of Pathology, Duke University, North Carolina, Durham, United States of America
[5] Department of Animal Science, National Chung Hsing University, Taichung, Taichung, Taiwan
[*] These authors contributed equally to this work.

## ABSTRACT

**Aims**. Studies have observed changes in autophagic flux in the adipose tissue of type 2 diabetes patients with obesity. However, the role of autophagy in obesity-induced insulin resistance is unclear. We propose to confirm the effect of a high-fat diet (HFD) on autophagy and insulin signaling transduction from adipose tissue to clarify whether altered autophagy-mediated HFD induces insulin resistance, and to elucidate the possible mechanisms in autophagy-regulated adipose insulin sensitivity.

**Methods**. Eight-week-old male C57BL/6 mice were fed with HFD to confirm the effect of HFD on autophagy and insulin signaling transduction from adipose tissue. Differentiated 3T3-L1 adipocytes were treated with 1.2 mM fatty acids (FAs) and 50 nM Bafilomycin A1 to determine the autophagic flux. 2.5 mg/kg body weight dose of Chloroquine (CQ) in PBS was locally injected into mouse epididymal adipose (10 and 24 h) and 40 $\mu$M of CQ to 3T3-L1 adipocytes for 24 h to evaluate the role of autophagy in insulin signaling transduction.

**Results**. The HFD treatment resulted in a significant increase in SQSTM1/p62, Rubicon expression, and C/EBP homologous protein (CHOP) expression, yet the insulin capability to induce Akt (Ser473) and GSK3$\beta$ (Ser9) phosphorylation were reduced. PHLPP1 and PTEN remain unchanged after CQ injection. In differentiated 3T3-L1 adipocytes treated with CQ, although the amount of phospho-Akt stimulated by insulin in the CQ-treated group was significantly lower, CHOP expressions and cleaved caspase-3 were increased and bafilomycin A1 induced less accumulation of LC3-II protein.

**Conclusion**. Long-term high-fat diet promotes insulin resistance, late-stage autophagy inhibition, ER stress, and apoptosis in adipose tissue. Autophagy suppression may not affect insulin signaling transduction *via* phosphatase expression but indirectly causes insulin resistance through ER stress or apoptosis.

Corresponding authors
Chih-Hsien Chiu,
chiuchihhsien@ntu.edu.tw
Yi-Fan Jiang, yfjiang@ntu.edu.tw,
F08644009@ntu.edu.tw

## INTRODUCTION

Although studies show that interaction of genetic and lifestyle factors cause type 2 diabetes (T2D), approximately 90% of people with T2D are overweight or obese (*Copps & White, 2012*). Adipocyte hypertrophy in obesity will induce the impairment of insulin sensitivity and the overexpression of some inflammatory cytokines that make the adipocyte dysfunctional. The pathway begins with the activation of the insulin receptor tyrosine kinase (IR) by insulin, which phosphorylates and recruits different substrate adaptors such as the insulin receptor substrate (IRS) family of proteins. Tyrosine phosphorylated IRS then displays binding sites for numerous signaling partners. Among them, PI3K has a significant role in insulin function, mainly *via* the activation of the Akt/PKB and the PKC cascades. Activated Akt induces glycogen synthesis by inhibiting GSK-3, protein synthesis *via* mTOR and downstream elements, and cell survival by inhibiting several pro-apoptotic agents. Akt phosphorylates and directly inhibits FoxO transcription factors. Inactivation of the receptor responses has been reported as the underlying cause of impaired insulin action (*Copps & White, 2012*). PH domain and leucine rich repeat protein phosphatase (PHLPP) directly dephosphorylates AKT at its hydrophobic motif (Ser473). Additionally, mutations in PTEN have been reported to cause insulin resistance and obesity (*Pal et al., 2012*). NHERF1 binds directly to PTEN and PHLPP1/2 *via* the PDZ domain and scaffolds ternary complexes at the membrane to suppress the activation of the PI3K–AKT pathway (*Molina et al., 2012*).

Recent studies have indicated extracellular disturbances, such as excess nutrient inflammation or hyperinsulinemia, cause intracellular stress in adipose tissues, which may damage these cells' ability to perform standard metabolic actions on insulin (*Fazakerley et al., 2019*). Concerning these intracellular stresses, some studies have identified autophagy as playing a vital role in regulation. *Yin et al. (2015)* showed that autophagy inhibition caused severe ER stress in adipocytes and thus induced adaptive responses to extracellular disturbances, attenuating insulin resistance deterioration (*Zhou et al., 2009*). Due to its ability to suppress insulin receptor signaling, ER stress has been proposed as an important factor in peripheral insulin resistance and type 2 diabetes (*Ozcan et al., 2004*).

During autophagosome biogenesis in macroautophagy, two ubiquitin-like systems, LC3 processing and Atg5-Atg12 conjugation, are associated with expanding the phagophore membrane (*Yang & Klionsky, 2009*). As for LC3, it is located on the membrane after post-translational modifications. The C-terminal end of the cytosolic form LC3-I is cleaved by Atg4, which is then activated by Atg7 and transferred to Atg3. Finally, a phosphatidylethanolamine (PE) will be conjugated to LC3-I to form LC3-II by Atg3 (*Kabeya et al., 2004*; *Hanada et al., 2007*). As for the Atg5-Atg12-Atg16 complex system, it promotes the formation of LC3-II (*Romanov et al., 2012*), which, in addition to participating in autophagosome formation, also has the function of recognizing autophagic cargos. LC3-II acts as a receptor to interact with the adaptor on the targets to promote their uptake and degradation. One of the best-characterized adaptor molecules is p62/SQSTM1, a multifunctional adaptor that enables the turnover of ubiquitinated substrates (*Glick, Barth & Macleod, 2010*). After expanding the phagophore membrane and cargo engulfment, the

mature autophagosome fuses with lysosome to form autolysosome for cargo degradation (*Mizushima, 2007*).

However, all of the above processes presuppose that the autophagy is still normal in the cells, but in the study of *Soussi et al. (2015)*, the autophagy in adipocytes from obese patients may be damaged. The connection between impaired autophagy and insulin resistance was observed by *Guo et al. (2017)*, who showed that Atg7 knockdown in 3T3-L1 adipocytes caused a reduction in the phosphorylation capacity of both the insulin receptor $\beta$ subunit and IRS-1 stimulated by insulin. As insulin resistance progresses, adipocyte autophagy is impaired in insulin-resistant states, in which type 2 diabetics demonstrate increased levels of autophagy-related proteins including ATG5, ATG7, BECN1, LC3-II, and LC3-I as well as autophagosome accumulation was observed in subcutaneous and visceral adipose tissue of non-diabetic obese and T2D patients as compared to lean individuals (*Rodríguez et al., 2012*; *Kosacka et al., 2015*). Moreover, *Cai et al. (2018)* also found that Akt was not activated by insulin in the white adipose tissue in an AdiAtg3KO mouse model, indicating that early-stage inhibition autophagy accelerated the decline in insulin sensitivity. However, the physiological function of damaged autophagy in adipose is unclear, including its role played in insulin resistance development. Therefore, this study was designed to confirm the effect of a high-fat diet (HFD) on autophagy and insulin signaling transduction from adipose tissue. We then clarified whether altered autophagy-mediated HFD induces insulin resistance. Afterward, we elucidated the possible mechanisms in autophagy-regulated adipose insulin sensitivity.

## MATERIALS & METHODS

### Animal

In this study, 8-week-old male C57BL/6 mice (obtained from the Laboratory Animal Center, College of Medicine, National Taiwan University) were housed at $25 \pm 2\ ^{\circ}C$, with an approximate 50–60% relative humidity and 12-hour light/12-hour dark cycle. Diets and water are freely accessed. Mice were acclimated for one month before the treatments started and placed in the cage according to their experimental group. Each cage contained 4–6 mice and was provided with enrichment (*i.e.*, plastic tube, shredded paper, *etc.*). The cage will be cleaned and checked every week. After the treatment, the mice were anesthetized with 2.5% avertin (0.15 mL/10 g) through intraperitoneal injection and sacrificed by cervical dislocation. All the operations on and the usage of animals followed the National Institutes of Health Guide for the Care and Use of Laboratory Animals (NIH Publications No.8023, advised 1978), and were approved by the National Taiwan University Institutional Animal Care and Use Committee (NTU105-EL-00178).

### High-fat diet-induced T2D in mice

B6 mice were randomly grouped to control diet (CTD) or high-fat diet (HFD) groups for 8 ($n = 4$) and 16 weeks ($n = 6$). The CTD and HFD were commercially available diets purchased from Research Diets (product numbers: D12450J and D12492). Before sacrifice at the 16th week, the mice were injected with insulin intraperitoneally for 30 min, and

blood samples were collected from the orbital sinus. After blood sample collection, mice were sacrificed by cervical dislocation for tissue harvest.

## Local chloroquine injection in epididymal adipose

The 12-week-old male C57BL/6 mice ($n = 4$) were anesthetized with 2.5% avertin (0.15 mL/10 g) through intraperitoneal injection, followed by surgery. The identified epididymal adipose tissues were injected with a 2.5 mg/kg body weight dose of chloroquine (Sigma-Aldrich) in phosphate buffered saline (PBS) or PBS alone as the control ($n = 6$). As a result of its suitability for *in vivo* study, chloroquine is used most frequently in mice to assess autophagic inhibition, as well as its affordability compared to Bafilomycin A1 (*Moulis & Vindis, 2017*). Mice were injected with insulin and sacrificed for tissue harvest at 10 and 24 h after surgery.

## Cell culture

3T3-L1 preadipocytes were purchased from Taiwan Bioresource Collection and Research Center (BCRC number: 60159). Cells were cultured in high glucose Dulbecco's modified eagle medium (DMEM) (D5648, Sigma-Aldrich; 12100046, Gibco) containing 10% newborn calf serum (16010159; New Zealand origin, Gibco), 1% penicillin-streptomycin (15140122; Gibco, Waltham, MA, USA) and 2.5 g/L of $NaHCO_3$. Cells were maintained at 37 °C with 5% CO2 supplement. For cell maturation, 3T3-L1 preadipocytes were switched to a medium that contained 10% fetal bovine serum (10270106; South American origin, Gibco) when the cells were 70% confluent. The medium was also added with 1 µM dexamethasone, 0.5 mM methyl isobutyl xanthine (IBMX), and 1 µg/mL insulin. After 2 days of treatment, the cells were maintained in insulin-only treatment until they were fully differentiated. 50 nM Bafilomycin (Tocris) and a dose of 2.5 mg/kg body weight of chloroquine were added to the culture medium for 4 and 24 h in 3T3-L1 mature adipocytes and treated insulin 15 min before sampling. A hydrophobic fluorescent dye Nile red was used to observe the oil droplet accumulation in 3T3-L1 mature adipocytes after the treatment. Additionally, we treated differentiated 3T3-L1 adipocytes with Bafilomycin A1 and FA. A mixture of 0.6 mM palmitic acid (PA) and 0.6 mM oleic acid (OA) was used to treat 3T3-L1 cells for 48 h. A mixture of OA and PA was dissolved in ethanol, and 1% BSA was added to the culture medium.

## Protein sample preparation and Western blotting analysis

Adipose tissues were homogenized in Pierce IP Lysis Buffer (87787, ThermoFisher Scientific) containing complete EDTA-free Protease Inhibitor Cocktail (04693132001; Roche, Basel, Switzerland) and PhosSTOP (04906845001; Roche), and then centrifuged at 4 °C, 16000×g for 20 min. Using Pierce BCA Protein Assay Kit (23225; Thermo Fisher Scientific, Waltham, MA, USA), the samples were diluted and mixed with 4X Laemmli Sample Buffer (1610747; Bio-Rad, Hercules, CA, USA) and 10 mM dithiothreitol (DTT) to give a final concentration of 2 µg/µL. As for the cell experiment, 3T3-L1 cells were rinsed once with PBS and collected directly in 1X Laemmli Sample Buffer plus 10 mM DTT after treatment. All harvested samples were immediately boiled at 98 °C for 10 min and then stored at −20 °C. Sodium dodecyl sulfate-polyacrylamide gel electrophoresis (SDS-PAGE)

was conducted to separate proteins with different molecular weights, depending on the target protein's size. Precision Plus Protein Dual Color Standards (1610374; Bio-Rad) and protein samples were loaded into gel wells. Electrophoresis was performed using running buffer (25 mM Tris, 192 mM glycine, 0.1% SDS), and the voltages were set at 95 V and 110 V according to the sample during the stacking and resolving stages.

After electrophoresis, the separated proteins were transferred onto methanol-pre-wetted and transfer buffer-pre-equilibrated PVDF membrane (1620177; Bio-Rad) *via* the transfer buffer (25 mM Tris, 192 mM glycine, 20% methanol). The protein size was determined by either Trans-Blot Turbo Semi-dry Transfer System (170-4155; Bio-Rad) or Criterion Wet Transfer Blotter (170-4070; Bio-Rad). The proteins were transferred at 25 V constant voltage with 0.4 A for 35 min for semi-dry transfer. For wet transfer, gels were equilibrated in transfer buffer for 15 min to remove excessive SDS, and then the proteins were transferred at 70 V with 250 mA for 1 h.

Polyvinylidene fluoride (PVDF) membranes were washed with methanol and stained with Ponceau S solution to visualize the transferred proteins. The fragment membranes cut at the target position were blocked with 5% non-fat milk in tris buffered saline with tween (TBST) and incubated with target protein-specific primary antibodies (Table 1), which were diluted with 1% BSA in TBST overnight at 4 °C. Membranes were washed with TBST three times for 15, 10, and 10 min each and then incubated with the species-specific horseradish peroxidase (HRP)-conjugated secondary antibodies (1:2500 to 1:5000 diluted with 5% non-fat milk in TBST) for 1 h at room temperature. Membranes were washed with TBST three times for 10, 5, and 5 min each, and then the blotting images were visualized with Bio-Rad (1705061) and GE (RPN2235) enhanced chemiluminescence (ECL) substrates reagents using a Bio-Rad ChemiDoc Touch Imaging System. Blotting quantification was performed using Bio-Rad Image Lab software. Table 1 presents details about the antibodies.

## RNA extraction

The RNA from mouse adipose tissues was extracted using TRIzol reagent (15596018; Thermo Fisher Scientific), which was homogenized at 4 °C. Afterward, the lysates were added to 100 µL of chloroform and then shook for 15 s. After being incubated for 2–3 min at room temperature, mixed lysates were centrifuged at $12,000 \times g$ for 15 min at 4 °C. Thereafter, the RNA-containing colorless upper aqueous phase was transferred to a new tube. The RNA was precipitated by adding 250 µL of isopropyl alcohol, mixing the solution for 10 min at room temperature, and then centrifuging it at $12,000 \times g$ for 10 min at 4 °C to spin down the RNA pellets. After centrifugation, the supernatant was removed, and the pellets were washed twice with 500 µL of 75% ethanol. Finally, the pellets were air-dried and dissolved in UltraPure DNase/RNase-Free Distilled Water (10977023; Thermo Fisher Scientific) by incubating them in a water bath at 60 °C. The RNA concentration was measured using a biophotometer.

## Reverse transcription and real-time PCR

Total RNA was reverse transcribed to cDNA using a PrimeScriptTM RT reagent Kit (RR037Q; TaKaRa). First, 200–500 ng RNA was mixed with PrimeScript buffer, PrimeScript

**Table 1 The antibodies used in the study.**

| Antibody name | Company product | Number |
|---|---|---|
| Anti-Akt | Cell Signaling Technology | 4691S |
| Anti-phospho-Akt Ser 473 | Cell Signaling Technology | 4060S |
| Anti-ATG5 | Cell Signaling Technology | 12994S |
| Anti-CHOP | Cell Signaling Technology | 2895S |
| Anti-cleaved caspase 3 | Cell Signaling Technology | 9664S |
| Anti-GAPDH | Cell Signaling Technology | 2118S |
| Anti-GSK-3 $\beta$ | Cell Signaling Technology | 3915S |
| Anti-phospho-GSK-3 $\beta$ S9 | Cell Signaling Technology | 5558S |
| Anti-LC3 | Cell Signaling Technology | 2775S |
| Anti-PHLPP1 | Merck Millipore | 07-1341 |
| Anti-PTEN | Cell Signaling Technology | 9559S |
| Anti-Rubicon | Cell Signaling Technology | 8465S |
| Anti-SQSTM1/p62 | Abcam | ab109012 |
| Goat anti-rabbit IgG-HRP | Santa Cruz Biotechnology | sc-2004 |

**Table 2 The primer pairs used in qPCR analysis.**

| Gene Name | Forward (5′ to 3′) | Reverse (5′ to 3′) |
|---|---|---|
| Map1lc3b | GGAGCTTTGAACAAAGAGTGGAA | GGTCAGGCACCAGGAACTTG |
| Actb | GTGCGTGACATCAAAGAG | CAAGAAGGAAGGCTGGAA |

RT enzyme mix I, 25 pmol oligo dT primer, and 50 pmol random 6 mers, and then it was incubated at 37 °C for 30 min followed by inactivation of reverse transcriptase at 85 °C for 10 s. The cDNA products were diluted with DNase/RNase-free distilled water to give a final concentration of 10–20 ng/μL, and then stored at −20 °C.

We used QuantStudio 3 System (Applied Biosystems, Foster City, CA, USA) for quantitative real-time PCR (qPCR) analysis to measure the level of target gene mRNA expression. A 10–20 ng of cDNA template was mixed with Fast SYBR Green Master Mix (Applied Biosystems) and 0.4 μM target gene-specific primer pairs in a total volume of 10 μL. The reaction mixtures were transferred to well plates and then centrifuged. The samples were denatured for 20 s at 95 °C followed by 40 cycles of the PCR stage, with the denaturing step at 95 °C for 3 s and the annealing and extension step at 60 °C for 30 s. Finally, a melting curve analysis was conducted. The expression levels of *Actb* were used as a loading control, and the gene-specific primer pairs are listed in Table 2.

## Serum free fatty acid content analysis

To compare serum free fatty acid levels of the CTD-fed and HFD-fed mice, we used a commercial assay kit: the Free Fatty Acid Quantification Kit from BioVision (K612-100). Fifty μL of serum samples were mixed with a 100 μL Reaction Mix. Then, the reactions were incubated at 37 °C for 30 min in the dark and the OD 570 nm value was measured. The sample readings were applied to the standard curve to obtain free fatty acid levels in the serum.
## Statistical analysis

Each experiment was replicated at least three times and data were expressed as mean ± standard error of the mean (SEM). Data were analyzed by Student's $t$-test or one-way ANOVA followed by the least significant difference test (LSD) with Statistics Analysis System (Version 9.4; SAS Institute Inc., Cary, NC, USA); $P < 0.05$ indicated statistically significant differences.

# RESULTS

## HFD induces autophagy impairment, insulin resistance, endoplasmic reticulum (ER) stress, and apoptosis in adipose tissue

The mice treated with 8 weeks of HFD significantly increased SQSTM1/p62 and Rubicon expression, but the expression levels of LC3-II and Atg5 were unaltered (Figs. 1A, 1B). Meanwhile, the mice treated with 16 weeks of HFD had a significantly increased expression of LC3-II and Atg5 (Figs. 1C, 1D), which suggests that the late stage of autophagy might have been inhibited and that the autophagosomes accumulated in a large amount to elevate the expression of LC3-II on the autophagosome membrane. Further, p62 was degraded primarily by autophagy, and if the lysosomal degradation of autophagosome is blocked, an accumulation of p62 is expected, and so the expression of Rubicon also significantly increased.

There was no significant difference in serum total (free fatty acids) FFAs between the HFD and CTD groups at 8 and 16 weeks from the blood sample (Fig. 1E). The western blotting data show that insulin injection might fail to induce Akt phosphorylation (Ser473) in adipose tissue from mice fed HFD for 16 weeks (Figs. 1F, 1G). Meanwhile, we also observed the marker of endoplasmic reticulum (ER) stress, C/EBP homologous protein (CHOP), was significantly increased. One of the key executioners of apoptosis, caspase-3, was activated as cleaved caspase-3 in adipose tissues from mice fed with HFD (Figs. 1H, 1I) (*Kadowaki & Nishitoh, 2013*; *Crowley & Waterhouse, 2016*).

We used differentiated 3T3-L1 adipocytes for FA treatment to reinforce the conclusion that autophagy was inhibited in the *in vivo* experiment. This was done by treating 3T3-L1 cells with a mixture of 0.6 mM palmitic acid (PA) and 0.6 mM oleic acid (OA) for 48 h. We observed that Bafilomycin A1 treatment given 4 h before sampling induces less expression of LC3-II protein in FA-treated cells than that in the BSA control group (Figs. 2A, 2B).

## Impaired late-stage autophagy may lead to insulin resistance indirectly by ER stress or apoptosis

To investigate whether autophagy inhibition can induce insulin resistance in a cell-autonomous manner, we conducted *in vitro* experiments using differentiated 3T3-L1 adipocytes. By treating differentiated 3T3-L1 with 40 μM of CQ for 24 h, we first addressed whether the autophagy in 3T3-L1 cells was inhibited. After treatment with CQ, we observed increased LC3-II and SQSTM1/p62 protein levels in 3T3-L1 cells (Figs. 3A, 3B, 3C). We found that hosphor-Akt (Ser473) in the CQ-treated group was significantly lower than that in the control (Figs. 3A, 3D).

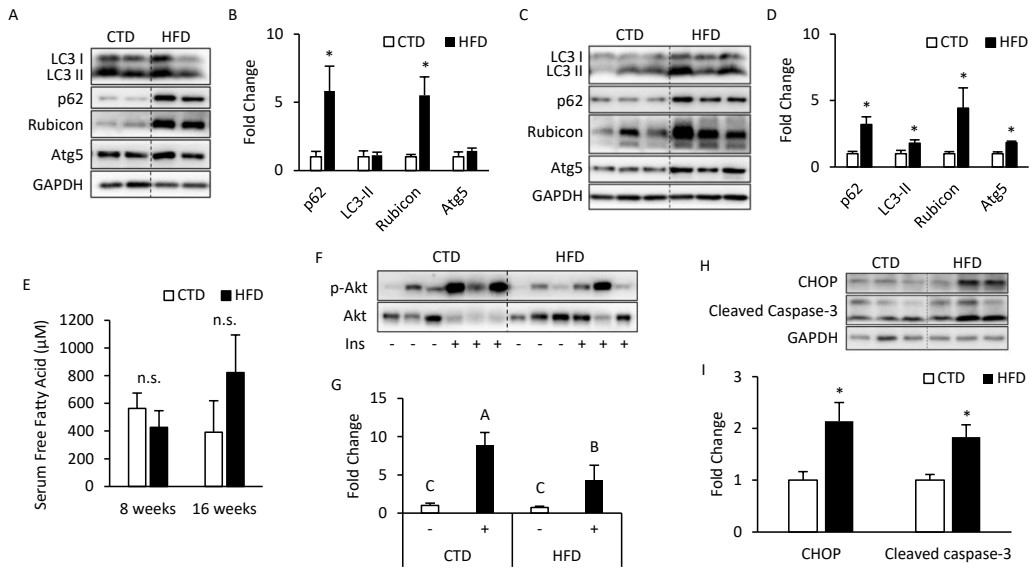

**Figure 1 Altered autophagy, insulin resistance, and ER stress in adipose tissues from mice fed with 8- and 16-week HFD.** The 12-week-old male B6 mice were fed with either the control diet or HFD for 8 weeks (A, B) and 16 weeks (C, D). T t he figure shows the immunoblots and the densitometric qua n tifi-cations of SQSTM1/p62, LC3, Rubicon, and Atg5 expression by Western blotting. Values are mean ± SEM ($n = 4$). The serum-free fatty acid levels between CTD-fed and HFD-fed mice were analyzed using a commercial assay kit (E). Values are mean ± SEM (8 weeks: $n = 4$ and 16 weeks: $n = 6$). n.s.: no significant difference. The phospho-Akt (Ser473) levels of adipose tissues from mice injected with or without insulin (0.5 IU/kg body weight I.P., 30 min) were analyzed at the 16th week using Western blotting (F, G). Values are mean ± SEM ($n = 6$). Different letters are considered statistically significant difference by one-way ANOVA and least significant difference (LSD) test, $P < 0.05$. Male B6 mice were fed with either the control diet or HFD for 16 weeks (H, I). The figure shows the immunoblots and the densitometric quantifications of C/EBP homologous protein (CHOP) and cleaved caspase-3 expression by Western blotting. Values are mean ± SEM ($n = 6$). An asterisk (*) indicates statistical significance, $P < 0.05$.

Interestingly, our Western blotting data showed that CQ-treated adipocytes had significantly higher CHOP levels and cleaved caspase-3 than those in the control (Figs. 3E, 3F), which echoed the *in vivo* experiment results observed in the adipose from mice fed with HFD. To further clarify whether insulin resistance was directly affected by autophagy inhibition or by concomitant ER stress or apoptosis, we excluded the effects of apoptosis on cells by reducing CQ concentrations. The protein levels in LC3-II in the treatment group were higher than those the control group, while the levels of SQSTM1/p62 were significantly increased at concentrations of 20 and 30 μM when compared to those of the control group (Figs. 4A–4C).

Moreover, the expression of cleaved caspase-3 did not change (Figs. 4A, 4D). Hence, we chose 20 μM dosage to determine if autophagy inhibition per se can induce insulin resistance; it did not influence insulin-stimulated Akt phosphorylation in 3T3-L (Figs. 4E, 4F).

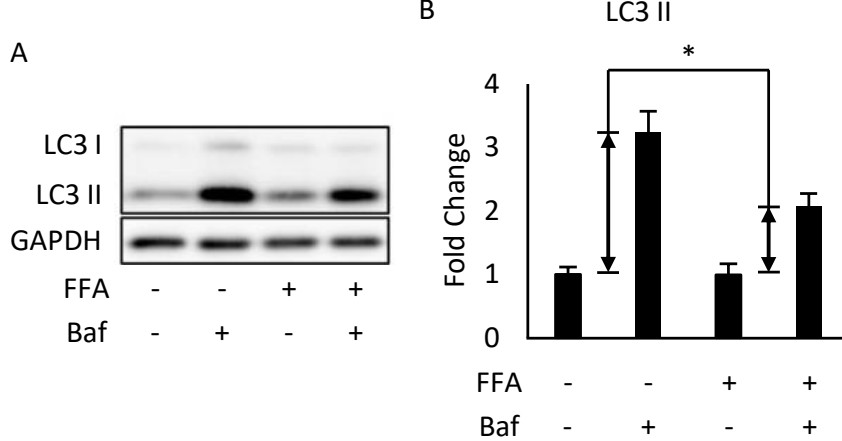

**Figure 2 Free fatty acids (FFAs) impaired the autophagic flux in differentiated 3T3-L1 adipocytes.** The Bafilomycin A1-induced LC3 II protein accumulation in 48-hour BSA- and FFA- (0.6 mM PA + 0.6 mM OA) loaded 3T3-L1 adipocytes were analyzed using Western blotting. The immunoblots (A), and calculated fold change of LC3 II protein levels were presented (B). For densitometric analyses, GAPDH was used as the loading control. Values are mean ± SEM ($n = 3$). An asterisk (*) indicates statistical significance, $P < 0.05$.

## The correlation between the late-stage autophagy inhibition with insulin resistance in adipose tissue

The results indicate that 10 h after CQ injection, the phosphorylation level induced by insulin was not significantly different from that of the control group (Figs. 5A, 5B). However, the capability of insulin to generate Akt (Ser473) and GSK3 $\beta$ (Ser9) phosphorylation was reduced 24 h after CQ injection (Figs. 5C–5E).

Additionally, although the protein levels of SQSTM1/p62 and LC3-II were no difference between CQ-treated and control groups (Figs. 6A, 6B), the LC3 mRNA expression was significantly decreased in adipose tissues with CQ injection (Fig. 6C). These data suggested that at the 24-hour post-injection of CQ, the suppression of late stage autophagy may have feedback effect which resulted in reduced mRNA level of LC3.

## The phosphorylation of insulin signaling blocked by autophagy is not associated with the expression of PHLPP1 and PTEN

After confirming that the inhibition of late-stage autophagy is related to the occurrence of insulin resistance, we further investigated the molecular mechanisms that may be involved. As it was previously reported that the expression of PHLPP1 is elevated in adipose tissue of obese patients and mice, which in turn leads to a decrease in phosphorylation of Akt and affected insulin sensitivity (*Andreozzi et al., 2011*), we first checked the protein level of PHLPP1 in mouse adipose treated with CQ. According to the Western blotting data, the CQ injection did not increase the expression of PHLPP1 compared to that in the control group. The amount of PTEN after CQ treatment was also not changed (Figs. 7A, 7B).

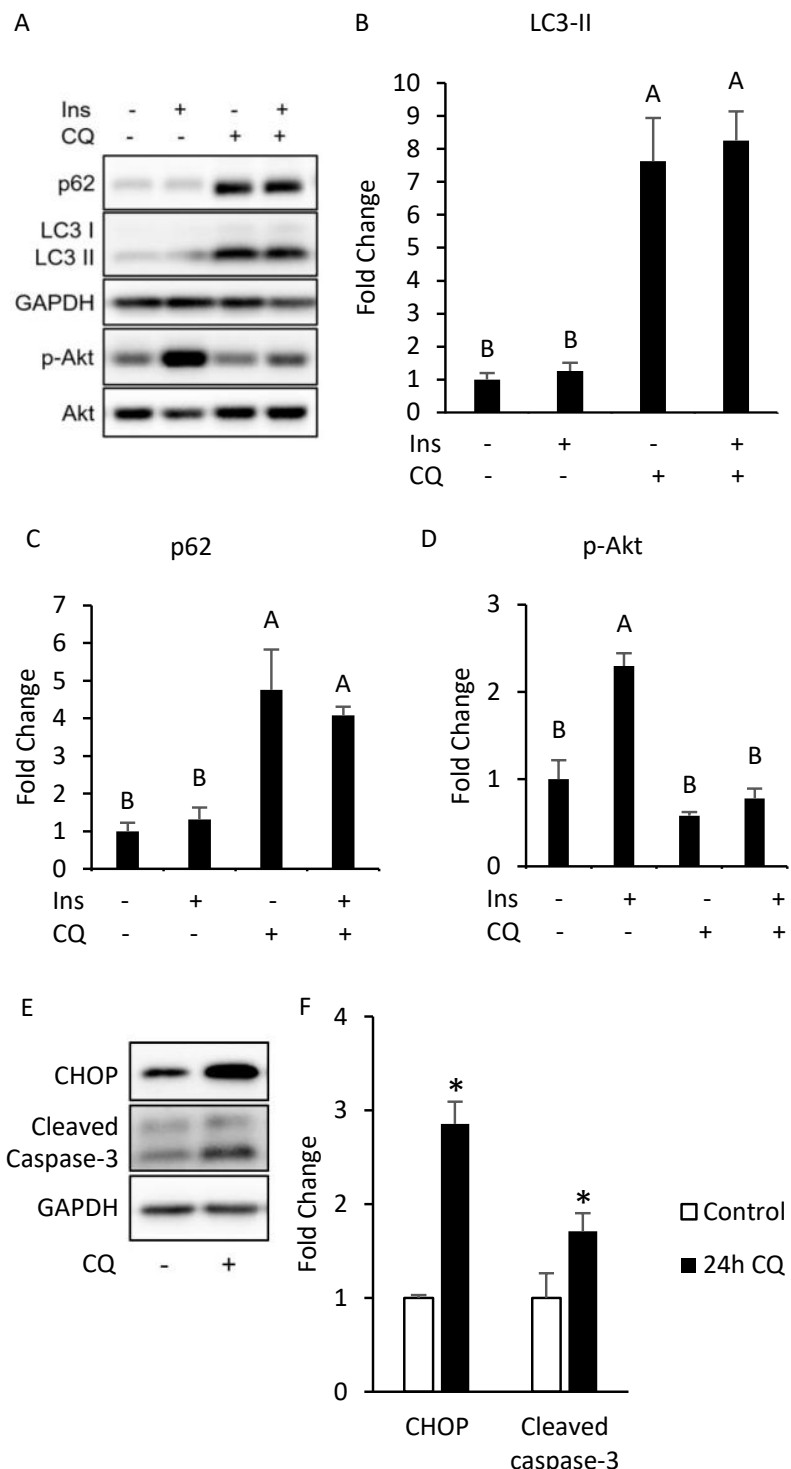

**Figure 3  Autophagy inhibition, insulin resistance, ER stress and apoptosis were found in 3T3-L1 after CQ treatment for 24 h.** The differentiated 3T3-L1 adipocyte was treated with 40 μM CQ for 24 h, and with 40 nM insulin for 15 min before sampling. SQSTM1/p62, (continued on next page...)

**Figure 3 (…continued)**
LC3 and p-Akt/Akt immunoblots (A) and densitometric quantifications (B, C, D). The differentiated 3T3-L1 adipocyte was treated with 40 μM CQ for 24 h. CHOP and cleaved caspase-3 immunoblot (E) and densitometric quantifications (F). For densitometric analyses of Western blotting data, GAPDH was used as the loading control. Values are mean ± SEM ($n = 3$). Different letters are considered statistically significant difference by one-way ANOVA and least significant difference (LSD) test, $P < 0.05$.

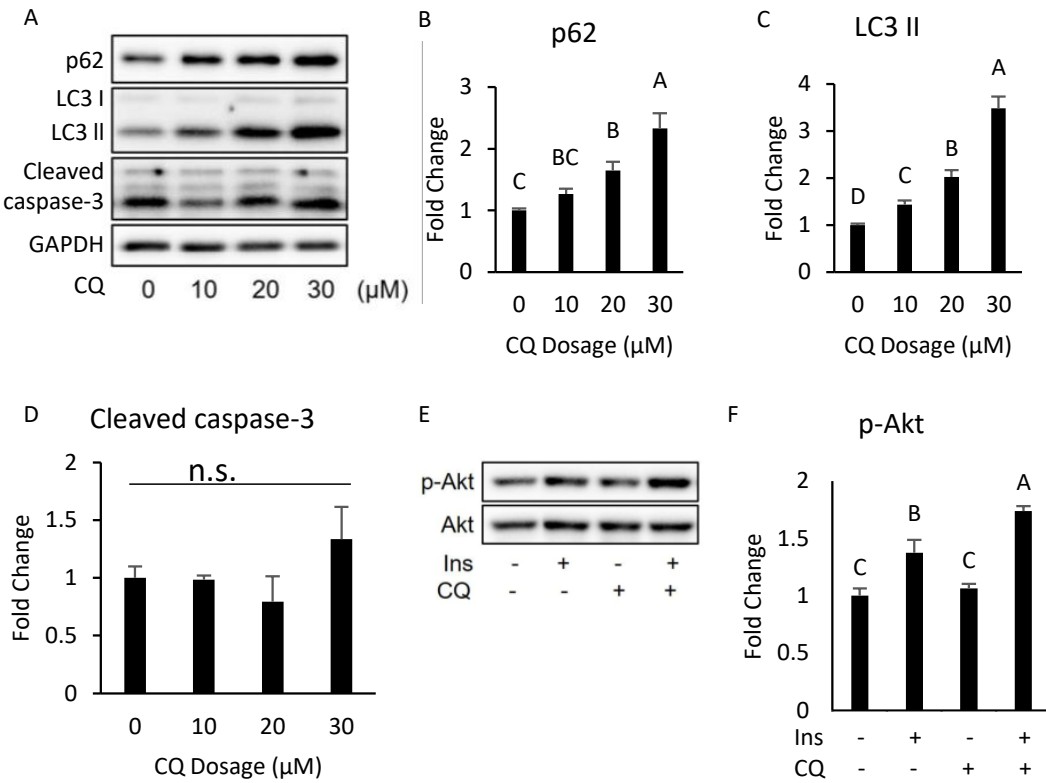

**Figure 4  Dose–response experiment of CQ treatment in 3T3-L1 cells and insulin resistance, ER stress, and apoptosis were observed in 3T3-L1 after 20 μM CQ treatment for 48 h.** The differentiated 3T3-L1 adipocyte was treated with 10, 20, and 30 μM CQ, for 24 h (A). The figure shows SQSTM1/p62 (B), LC3 (C), and cleaved caspase-3 (D) immunoblots and densitometric quantifications. The differentiated 3T3-L1 adipocyte was treated with 20 μM CQ for 24 h and with 40 nM insulin for 15 min before sampling. The phospho-Akt (Ser473) and total Akt levels of 3T3-L1 adipocyte were analyzed using Western blotting (E, F). GAPDH was used as the loading control. Values are mean ± SEM ($n = 3$). Different letters are considered statistically significant difference by one-way ANOVA and least significant difference (LSD) test, $P < 0.05$. n.s.: no significant difference.

## DISCUSSION

Insulin resistance is considered one of the crucial factors in early-stage T2D development, which can cause metabolic dysfunction in tissues. As cellular functions that may be involved in insulin resistance, studies have pointed out that autophagy plays a critical role in the progression of T2D, which is related to the adaptive response to intracellular stress. In particular, *Cai et al. (2018)* observed that autophagy ablation can damage the

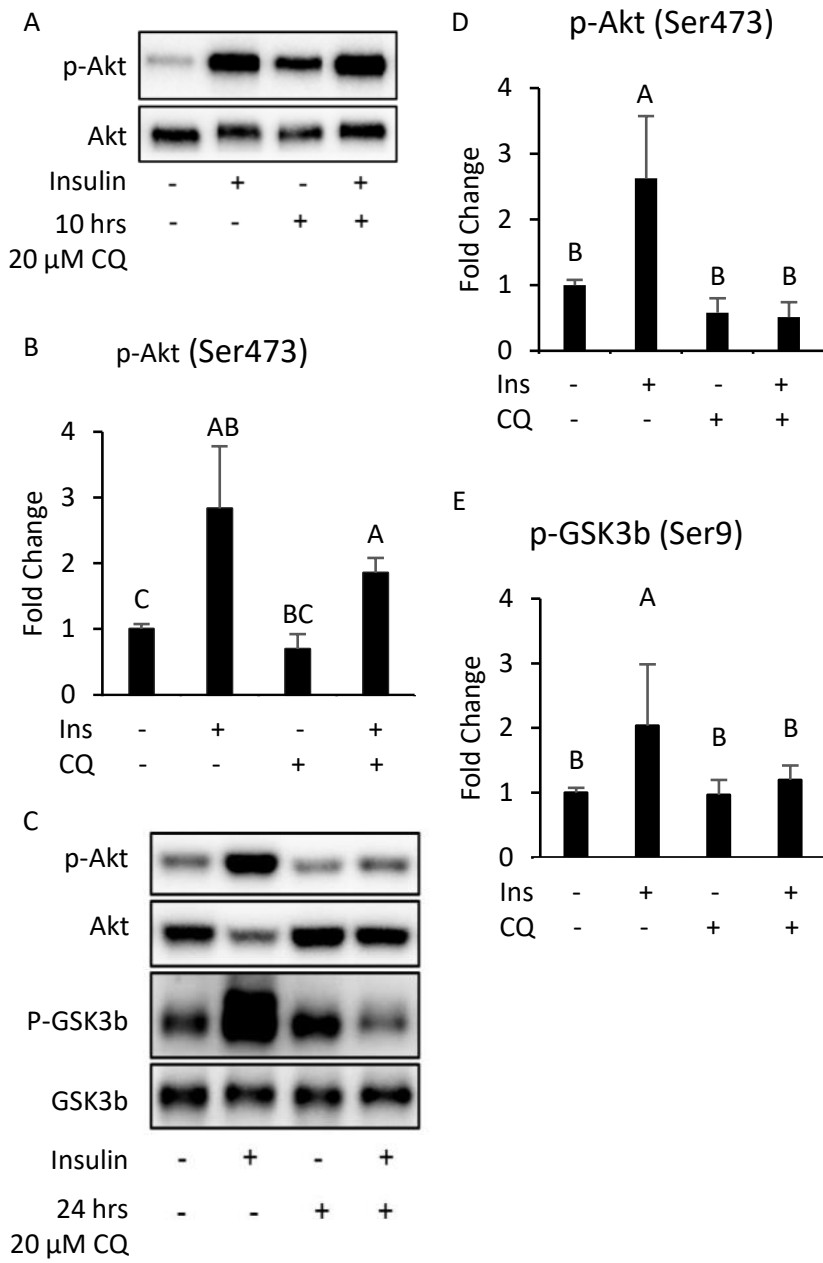

**Figure 5** **The insulin signaling pathway after chloroquine (CQ) treatment for 10 h and 24 h.** B6 mice were injected with PBS or CQ into epididymal adipose tissue for 10 h (A, B) and 24 h (C, D, E) mice injected with or without I.P. insulin (0.5 IU/kg body weight, 30 min) before sacrifice. The phospho-Akt (Ser473) and total Akt levels of adipose tissues were analyzed with Western blotting. Blotting and quantitative data of p-Akt/Akt and p-GSK3 $\beta$/ GSK3 $\beta$ were presented. Values are mean $\pm$ SEM ($n = 4$) (A) ($n = 6$) (C). Different letters are considered statistically significant difference by one-way ANOVA and least significant difference (LSD) test, $P < 0.05$. GAPDH was used as the loading control. The LC3B mRNA level in 24-hour PBS- and CQ-treated adipose was measured using qPCR analysis. For data quantification, the housekeeping gene ACTB was used as the internal control. Values are mean $\pm$ SEM ($n = 6$). An asterisk (*) indicates statistical significance, $P < 0.05$. n.s.: no significant difference.

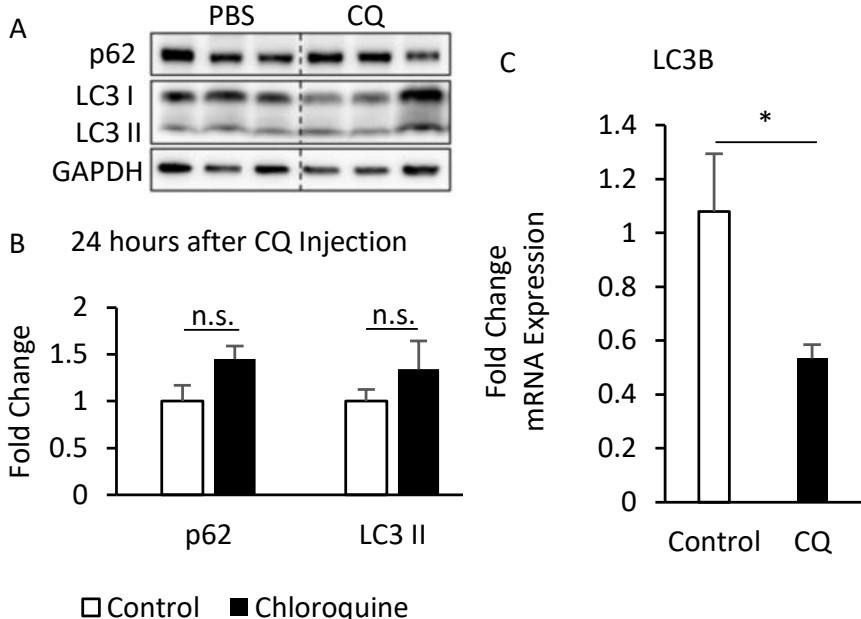

**Figure 6 The suppression of late-stage autophagy may have feedback effect which resulted in reduced mRNA level of LC3B.** B6 mice were injected with PBS or CQ into epididymal adipose tissue for 24 h. Representative SQSTM1/p62 and LC3 immunoblots and densitometric quantifications (A, B). For densitometric analyses of Western blotting data, GAPDH was used as loading control. The LC3B mRNA level in 24-hour PBS- and CQ-treated adipose was measured using qPCR analysis (C). For data quantification, the housekeeping gene ACTB was used as the internal control. Values are mean ± SEM ($n = 6$). An asterisk (*) indicates statistical significance, $P < 0.05$. n.s.: no significant difference.

insulin signaling transduction in adipose tissues. However, the role of autophagy in adipose function and the molecular mechanisms of how autophagy inhibition affects the insulin signaling pathway remain unclear.

HFD treatment for 8 weeks led to a blockade of late-stage autophagy in adipose, and the inhibition of autophagy was observed even at the 16th week (Figs. 1B, 1D). Rubicon's increased expression was the prominent cause of suppression at the late stage, which was similar to the results of *Tanaka et al. (2016)* and *Wang et al. (2017)* in hepatocytes. Fatty acid-treated cells showed a significant decrease in autophagic flux (Fig. 2), which doubly confirmed that autophagy was inhibited and is also consistent with the clinical findings of *Soussi et al. (2015)* Meanwhile, endoplasmic reticulum (ER) stress and apoptosis were also identified in adipose tissue (Fig. 1I). Studies have demonstrated that these are two effects of HFD, and *Kawasaki et al. (2012)* further clarified that ER stress is induced by reactive oxygen species (ROS) generation and inflammatory cytokines. However, *Feng et al. (2011)* showed that the inflammatory response does not cause apoptosis. Our study brings new information that HFD might cause late-stage autophagy suppression in adipose associated with Rubicon upregulation. At the same time, a series of complex reactions, including insulin resistance, ER stress, and apoptosis, also occur.

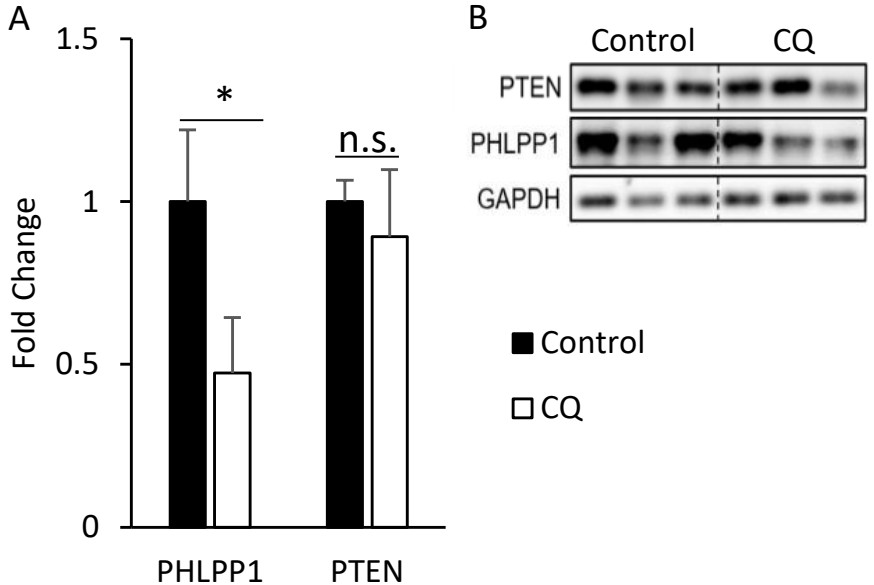

**Figure 7 Autophagy may regulate insulin-stimulated signal transduction without associated with PHLPP1 and PTEN.** The PHLPP1 and PTEN level of PBS- and CQ-injected mouse adipose via Western blotting. An asterisk (*) indicates statistical significance, $P < 0.05$. Quantitative data (A) and Western blotting (B) of PTEN were presented. For densitometric analyses of Western blotting data, GAPDH was used as the loading control. Values are mean ± SEM ($n = 3$). n.s.: no significant difference.

The suppression of late-stage autophagy may damage insulin signaling transduction (Fig. 5D). This might be associated with the time point of sampling, or that the local injection did not allow the drug to spread throughout the tissue. Interestingly, the mRNA expression of LC3 in early-stage autophagy was reduced (Fig. 6C), thus suggesting that the inhibition of late-stage autophagy may cause feedback to decrease the mRNA level of LC3 and subsequently block the early-stage autophagy. According to previous studies, if the early-stage autophagy is sometimes suppressed, insulin resistance may worsen (*Guo et al., 2017*; *Cai et al., 2018*).

We hypothesized that some phosphatases could not be degraded because of the blocked autophagic degradation, which would increase their expression and hinder the cascade of phosphorylation events in the insulin signaling pathway. The protein levels of PH domain and leucine-rich repeat protein phosphatase 1 (PHLPP1) and protein-tyrosine phosphatase 1B (PTP1B), in adipose from obese mice or patients exhibit a significant increase compared to lean animals (*Andreozzi et al., 2011*; *Zabolotny et al., 2008*). In this study, the amount of PHLPP1 did not elevate as expected in the CQ-treated group but instead decreased significantly (Fig. 7A). It shows that autophagy suppression may not directly cause Akt phosphorylation to be restricted due to increased expression of PHLPP1; conversely, it reduced the protein level of PHLPP1. In this regard, we suppose that autophagy which mediated the degradation of Mir6981 was inhibited, thereby leading to the abundance of Mir6981 that mitigated the protein translation of PHLPP1 (*Peng et al., 2019*). In addition, there was no significant difference in phosphatase and tensin homolog (PTEN) between

the CQ-treated and control groups (Fig. 7A). This indicates that PTEN may also not be involved in the insulin signaling transduction blocked by autophagy inhibition. In summary, autophagy inhibition at late stages blocked insulin signaling downstream independent of PHLPP1 and PTEN. However, more experiments are required to establish this.

At a high dose (40 μM) of CQ, the insulin sensitivity in 3T3-L1 adipocytes was reduced and accompanied by ER stress and apoptosis (Fig. 3), which was almost identical to the findings from the *in vivo* experiments. However, insulin signaling was still transduced normally in 3T3-L1 treated with CQ, thus suggesting that autophagy suppression may not directly contribute to insulin resistance. However, it is also possible that the abnormality of the insulin signaling pathway did not occur at the time of sampling. After CQ treatment for 24 h, the 3T3-L1 adipocytes developed insulin resistance as expected, but ER stress and apoptosis were also induced (Fig. 3F). These results demonstrate that inhibition of late-stage autophagy per se did not directly cause insulin resistance in the adipocyte. Studies have indicated that CQ blocks the fusion of autophagosome and lysosome, which leads to the accumulation of a large number of damaged proteins in the cytoplasm and induces ER stress.; persistent ER stress eventually results in cell death (*Jia et al., 2018*). Moreover, *van der Kallen et al. (2009)* showed that the relation of ER stress with insulin resistance is more evident than its relation with apoptosis. In adipose, ER stress triggers activation of c-Jun N-terminal kinase (JNK) through activated inositol requiring 1 alpha (IRE1 $\alpha$), thereby inhibiting serine phosphorylation of IRS1, and resulting in insulin resistance (*Zhang & Kaufman, 2008*). In addition, adipocyte apoptosis contributes to macrophage infiltration into adipose tissues, which increases the production of inflammatory cytokines. And these cytokines can affect the insulin signaling, such as TNF $\alpha$ which also inhibits IRS1 phosphorylation (*Alkhouri et al., 2010*). In summary, the suppression of late-stage autophagy caused by HFD may lead to a large accumulation of impaired proteins in adipocytes, which in turn leads to ER stress and apoptosis. And the latter two will eventually induce insulin resistance in adipose tissue.

## CONCLUSIONS

A long-term high-fat diet promotes insulin resistance in adipose tissue and leads to the increased protein levels of Rubicon, which blocks late-stage autophagy and is accompanied by endoplasmic reticulum (ER) stress and apoptosis. Among these conditions, inhibition of late-stage autophagy is associated with a decrease in insulin sensitivity. However, autophagy suppression may not affect the insulin signaling transduction *via* the pathway of PHLPP1 and PTEN but instead causes insulin resistance indirectly through ER stress or the apoptosis pathway.

## ACKNOWLEDGEMENTS

We thank Nigel Daly for his diligent proofreading of this article.

### Funding

This study was supported by grants from the Ministry of Science and Technology (109-2320-B-002 -038 -MY3) (to Yi-Fan Jiang, National Taiwan University) and (106-2320-B-002-040-MY3 and 109-2314-B-002-099-MY3) (to Chih-Hsien Chiu, National Taiwan University). The funders had no role in study design, data collection and analysis, decision to publish, or preparation of the manuscript.

### Grant Disclosures

The following grant information was disclosed by the authors:
Ministry of Science and Technology:  109-2320-B-002 -038 -MY3.
National Taiwan University: 106-2320-B-002-040-MY3,  109-2314-B-002-099-MY3.

### Competing Interests

The authors declare there are no competing interests.

### Author Contributions

- Yovita Permata Budi performed the experiments, analyzed the data, prepared figures and/or tables, authored or reviewed drafts of the article, and approved the final draft.
- Yi-Hsuan Li conceived and designed the experiments, performed the experiments, analyzed the data, prepared figures and/or tables, authored or reviewed drafts of the article, and approved the final draft.
- Chien Huang performed the experiments, authored or reviewed drafts of the article, and approved the final draft.
- Mu-En Wang performed the experiments, authored or reviewed drafts of the article, and approved the final draft.
- Yi-Chun Lin conceived and designed the experiments, authored or reviewed drafts of the article, and approved the final draft.
- De-Shien Jong conceived and designed the experiments, authored or reviewed drafts of the article, and approved the final draft.
- Chih-Hsien Chiu conceived and designed the experiments, authored or reviewed drafts of the article, and approved the final draft.
- Yi-Fan Jiang conceived and designed the experiments, prepared figures and/or tables, authored or reviewed drafts of the article, and approved the final draft.

### Animal Ethics

The following information was supplied relating to ethical approvals (i.e., approving body and any reference numbers):
   The Institutional Animal Care and Use Committee (IACUC) approved the study with the IACUC Approval No: NTU105-EL-00178.

### Data Availability

   Raw data is available in the Supplementary Files.

## Supplemental Information

Supplemental information for this article can be found online at http://dx.doi.org/10.7717/peerj.13867#supplemental-information.

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
