# Peer review of "The role of autophagy in high-fat diet-induced insulin resistance of adipose tissues in mice"

_PeerJ, doi:10.7717/peerj.13867_

## Round 0.1 · original submission · Major Revisions

· Academic Editor

Major Revisions

Thanks for submitting this interesting work to PeerJ. All reviewers and I agree that there is merit in this study. All reviewers have raised some points for you to address. These are largely minor issues of style and presentation which should be easy for you to address.

More substantive issues which will require careful work are to make sure the results and figures match (see reviewer-1) and carefully edit the figure legends. Also, please tone down the conclusions - as reviewer-3 notes you are over-reaching.

Please make clear in ALL figure legends how many technical and biological replicates were performed.

Finally, please note that all experimental data should be presented, so for three biological replicates of immunoblots, upload all three blots for each antibody etc.

Reviewer 1 ·

Basic reporting

In this study, the author demonstrated that high-fat diet treatment caused the suppression of late-stage autophagy, endoplasmic reticulum (ER) stress, and apoptosis in the adipose tissue of mice. The author also indicated that autophagy inhibition didn’t change the insulin signaling transduction but caused insulin resistance indirectly through ER stress or apoptosis.
This studying is meaningful. The manuscript is well written in English. The structure of the manuscript conforms to PeerJ standards. Raw data and supporting information were supplied. 
However, PeerJ uses the 'Name. Year' style with an alphabetized reference list. Please format the in-text citations and reference section following the PeerJ guidance.

Experimental design

The experimental design is clear. The method is writing in detail. Here are two minor points that may need to be addressed.
1. Line 25: What is the Bafilomycin A1 working concentration?
2. Line 220-222: How to treat the cells with FFAs is suggested to be included in the Materials & Methods section.

Validity of the findings

The result section is not very clear. Some results are not matched with the figures. Some figure legends are not consistent with the pictures. That caused big confusion. Please rearrange these figures. I have some comments as follows:
1. In Figure 1F, Figure 5 E, Figure 6A, and Figure 6C, the general housekeeping reference protein should be included.
2. In Figure 7, there is no A or B labeled.
3. Figure 3 legend is not matched with Figure 3.
4. Line 230-231: “After treatment with CQ, we observed increased LC3II and SQSTM1/p62 protein levels in 3T3-L1 cells (Figures 3B, 3C). ” But Figure 3B showed LC3II and SQSTM1/p62 protein levels were no differences between the control group and CQ treatment group. In Figure 3C, the LC3B mRNA level was decreased in the CQ treatment group. The result is not clear here.
5. Line 232-234: lack of Figure 3D and Figure 3F.
6. Figure 4 legend is not matched with Figure 4.
7. Figure 5 legend is not matched with Figure 5. 
8. Figure 6 legend is not matched with Figure 6.

Additional comments

NO comments

Reviewer 2 ·

Basic reporting

No comment.

Experimental design

No comment.

Validity of the findings

No comment.

Additional comments

In the present study, Yovita Permata Budi and colleagues evaluated changes in autophagy and insulin signaling transduction in adipose tissue from C57BL/6 mice fed a high-fat diet (HFD) in order to elucidate potential mechanisms in autophagy-regulated adipose insulin sensitivity. The authors found that HFD induced late-stage autophagy inhibition, endoplasmic reticulum (ER) stress and apoptosis in the adipose tissue. Autophagy suppression did not influence insulin signalling via phosphatase expression, but indirectly caused insulin resistance through ER stress or apoptosis. This is an interesting study shedding light into the communication of autophagy and insulin resistance in the context of obesity. Nonetheless, some specific points require to be amended.

Specific comments

1. Title: please change “diet induced” with “diet-induced”.
2. Abstract: please include age and sex of experimental animals as well as the dose and time for chloroquine administration.
3. Introduction, lines 62-63: this sentence is confusing since ER stress has been proposed as a central feature of peripheral insulin resistance and type 2 diabetes due to its ability to supress insulin receptor signalling (Ozcan U et al. Science 2004, PMID: 15486293)
4. Introduction, line 77-78: In the study of Sossi et al. the authors state that adipocyte autophagy is attenuated in human obesity. However, it is important to explain that this condition conversely progresses to increased expression of adipocyte autophagy flux in insulin-resistant states, with patients with type 2 diabetes showing increased expression of autophagy-related proteins (ATG5, ATG7, Beclin-1 and LC3C) and autophagosome accumulation than induviduals with normal weight or with obesity and normoglycemia (Rodríguez A et al. DIabetologia 2012, PMID: 22869322; Kosacka J et al. Mol Cel Endocrinol 2015, PMID: 25818883).
5. Methods, line 154: please include the reference to Table 1 after “all target protein-specific primary antibodies”,
6. Murine genes should be written according to the approved nomenclature. Murine gene symbols should be written in italics, with the first letter capitalised, without hyphens and greek letters (e.g. Actb for the gene encoding -actin).
7. Please avoid informal abbreviations, such as doesn’t, didn’t, haven’t, throughout the manuscript.

·

Basic reporting

The manuscript accomplishes all the essential criteria, meeting all the standards.
1.-Structure, figures and tables meet the criteria and show the raw data.
2.- They include the custom checks.
3.- All the uncropped westerns are also included.
4.- English seem to be well used (I am not an English native person).
5.- Literature references are OK.

Experimental design

They showed a correct experimental design.

Validity of the findings

They show robust results but I will tone down a bit the discussion and conclusions.

Additional comments

Dear authors,

Your manuscript is very well performed, it is thorough research about the HFD effect on Type II diabetes. However, a few details may be improved:
1.- In the in vivo assays, it would be interesting to know the inflammation after the surgery, which can be affected by the procedure and affects the autophagy results.
2.- Figures need to be checked as the letters of the legends don't correspond to the figures explained.
3.- Altered autophagy is shown already at 8 months, reanalyse the flux results.
4.- I would recommend testing the toxicity of the doses of the fatty acid, which seems a bit high.
5.- State the reason for using chloroquine instead of leupeptin or another lysosomal inhibitor in vivo.
6.- Tone down the discussion and the conclusions, they are a bit overreaching.

---

## Round 0.2 · Minor Revisions

· Academic Editor

Minor Revisions

Thank you for carefully addressing all the comments; one reviewer has identified a couple of minor points which remain outstanding. Could you address these please? It will not be necessary to send out for further review after this.

Reviewer 1 ·

Basic reporting

No comments

Experimental design

No comments

Validity of the findings

No comments

Additional comments

The majority of the reviewer’s questions have been addressed. However, there are some minor points that should be addressed.
1. Line 225: The result should start from Figure 1A. All the Figures (A, B, C...) should be included in the context. Please check all the result section.
2. Figure 7 legend: “Western blotting (A) and quantitative data (B) of PTEN were presented” However, quantitative data of PHLPP1 and PTEN is A, and WB is B in this Figure. In Figure 7A, the author used the “control and CQ”, but in Figure 7B, the WB was labeled by PBS and CQ. Please correct it and make the labeling consistent.

Reviewer 2 ·

Basic reporting

I thank the authors for addressing all my comments/suggestions. The manuscript has been substantially improved, and it is my belief that it is now ready for publication.

Experimental design

The experimental design is well-defined and relevant, and the methodology is clearly explaines in the revised manuscript.

Validity of the findings

The results are clearly stated and robust, and the conclusion is linked to the original research question.

---

## Round 0.3 · accepted · Accept

· Academic Editor

Accept

Thanks for attending to the remaining minor issues. Congratulation's on a nice study.